# Microbial Metabolite Regulation of Epithelial Cell-Cell Interactions and Barrier Function

**DOI:** 10.3390/cells11060944

**Published:** 2022-03-10

**Authors:** Alfredo Ornelas, Alexander S. Dowdell, J. Scott Lee, Sean P. Colgan

**Affiliations:** 1Mucosal Inflammation Program, Department of Medicine, University of Colorado Anschutz Medical Campus, 12700 E. 19th Ave, Mailstop B146, Aurora, CO 80045, USA; alfredo.ornelassanchez@cuanschutz.edu (A.O.); alexander.dowdell@cuanschutz.edu (A.S.D.); joseph.s.lee@cuanschutz.edu (J.S.L.); 2Rocky Mountain Regional Veterans Affairs Medical Center, 1700 N. Wheeling St., Aurora, CO 80045, USA

**Keywords:** bacterial metabolites, intestinal mucosal barrier, intestinal epithelial cells, tight junctions, inflammation, SCFAs, indoles, hypoxanthine, secondary bile acids, polyamines

## Abstract

Epithelial cells that line tissues such as the intestine serve as the primary barrier to the outside world. Epithelia provide selective permeability in the presence of a large constellation of microbes, termed the microbiota. Recent studies have revealed that the symbiotic relationship between the healthy host and the microbiota includes the regulation of cell–cell interactions at the level of epithelial tight junctions. The most recent findings have identified multiple microbial-derived metabolites that influence intracellular signaling pathways which elicit activities at the epithelial apical junction complex. Here, we review recent findings that place microbiota-derived metabolites as primary regulators of epithelial cell–cell interactions and ultimately mucosal permeability in health and disease.

## 1. Introduction

The gastrointestinal (GI) tract is lined by a monolayer of epithelial cells that form a critical barrier separating our body from the external environment. In this, the intestinal epithelium plays an essential role in the regulation of the immune system and maintenance of health [1]. The GI tract performs two key tasks: it acts as a highly selective filter, allowing movement of nutrients and water into circulation and the internal milieu in general. As a barrier, it prevents exposure to harmful entities and restricts infiltration of pathogenic organisms [2]. This seemingly conflicting task is finely regulated by an interplay of structural components and molecular interactions at the intestinal mucosa to maintain intestinal integrity and immune homeostasis [3]. The function of the mucosal barrier can be affected through severe structural damage or more subtle changes in its regulating components [4]. Although the exact mechanisms are still unclear, loss of barrier function is tightly linked to chronic inflammatory diseases, such as inflammatory bowel disease (IBD) [5].

The intestine is home to the largest constellation of microorganisms in our body including bacteria, viruses and fungi termed the gut microbiota (GM), mostly present in the colon. The fact that the intestine is not easily overtaken by infection is a testament to the monumental task accomplished by the mucosal barrier. A recently revised study revealed that in the body, the ratio of human:bacterial cells is close to 1:1 [6]. Interestingly, the human genome consists of approximately 23,000 genes, whereas the microbiome encodes over 3 million genes [7]. Therefore, the “human superorganism” is actually approximately 1% human in this regard. These microbial passengers in return offer many symbiotic benefits to the host in the form of thousands of metabolites, acting upon a range of physiological functions including strengthening gut integrity, modulating cellular energetics and shaping the intestinal epithelium [8]. There is a wide variety of metabolites necessary for maintaining and restoring barrier function including short-chain fatty acids (SCFAs), indoles, purines, bile acids, and polyamines. This review will focus on the role of microbial-derived metabolites in the regulation of intestinal barrier function and homeostasis.

## 2. Intestinal Barrier Composition

The intestinal barrier structure is composed of several elements that, in conjunction, provide this complex physical and immunological defense mechanism. The first line of defense is an outer mucus barrier, where the commensal bacteria, antimicrobial peptides (AMPs), and secretory immunoglobulin A (sIgA) reside; next, a central monolayer of specialized intestinal epithelial cells (IECs) are tightly held together by tight junctions (TJs), adherens junctions (AJs), and desmosomes; lastly, the inner lamina propria, where the last defense resort of innate and adaptive immune cells reside, such as T cells, B cells, macrophages, and dendritic cells [1,9].

Bacteria arriving at the gut lumen are immediately met by the mucus barrier, which inhibits their direct contact with the intestinal epithelium [9]. This layer of protection is composed of a hydrated gel of heavily glycosylated proteins termed mucins. In the intestinal mucosa, mucin 2 (MUC2) is the most abundant mucus protein secreted by goblet cells [1]. MUC2 is the primary component of the mucus barrier in the colon and is critical for protection against disease, as demonstrated in a classical model of MUC2 knockout mice spontaneously developing colitis [10]. Furthermore, transmembrane mucins expressed by IECs remain attached to their apical surface and, together with glycolipids, form the glycocalyx [11]. Composed of various receptors for bacterial adhesion, a critical function of the glycocalyx is to act as a landing hub for normal microbiota and limit colonization by pathogens [12,13,14,15]. Remarkably, the mucus in the colon is organized in two layers: an inner layer that is “firmly” adherent to epithelial cells and an outer non-attached “soluble” layer. While the inner layer is dense and does not allow bacteria to penetrate, thus keeping the epithelial cell surface free of bacteria, the outer layer plays a crucial role as a proliferating habitat of the microbiota in the colon [16]. Lastly, there is the presence of immune regulators such as AMPs and sIgA molecules as final reinforcement in the physical separation of the epithelium to the bacteria-invaded lumen [17]. AMPs have the capacity to rapidly kill [18] or arrest [19] a wide variety of microorganisms. An important class of AMPs are defensins, which are broadly classified as α and β defensins. Alpha-enteric defensins (HD5 and HD6) are secreted by Paneth cells and act as antimicrobial agents in the small intestine [20,21]. Secretion of these peptides is upregulated in the colonic mucosa of IBD patients through metaplastic Paneth cells [22]. Additionally, β-defensins secreted by the colonic epithelium play a major role in innate host defenses to maintain a healthy microbiota. Accordingly, their important role is reflected in IBD patients, where the defective expression and function of β-defensins lead to altered microbiota, infection and inflammation as reviewed here [23]. To briefly elucidate their antimicrobial mechanisms, defensins are cationic and arginine-rich peptides that bind to negatively charged microbial membranes, causing membrane disruption and cell death [24]. An interesting alternative mechanism was reported in HD6, where the peptide spontaneously self assembles into multi-peptide nanonets in the presence of bacteria [19]. In other words, rather than killing bacteria, HD6 function by aggregating and sequestering bacteria. Notably, the mucosal layer and the gut microbiota are interdependent: a shift in the microbiota will affect the composition of the mucosa and vice versa [25].

Epithelial cells provide the most selective component of the intestinal physical barrier. A pool of pluripotent stem cells residing in the crypt produce a distinct type of cells including absorptive enterocytes, goblet cells, enteroendocrine cells, Paneth cells, and microfold cells [26]. These cells, in conjunction, form a continuous and polarized monolayer barrier that separates the lamina propria from the lumen. The epithelial monolayer allows for selective passage of water, nutrients, and electrolytes while excluding harmful microbial pathogens, toxins, and other foreign agents [27]. This selective filter is regulated by the presence of junctional complexes. The three main complexes are the TJs, AJs, and desmosomes [28]. TJs are located in the apical side of the epithelial layer and form a continuous belt-like ring between the apical and lateral membrane [29]. TJs consist of transmembrane proteins (e.g., occludin and claudin), peripheral membrane proteins (e.g., zonula occludens ZO-1 and ZO-2), and regulatory proteins. TJ proteins regulate molecule passage through the epithelial layer based on their size and charge. They are essential to maintain a strong epithelial barrier and gut health. AJs and desmosomes are found below TJs and provide strong mechanical attachments between neighboring cells [29]. The structures, properties, and functions of these complexes have been extensively reviewed elsewhere [30,31]. The disruption of these complexes leads to decreased barrier and immediate invasion of inflammatory agents. Continuous inflammation perpetuates the physical impairment of their function and can lead to further aggravation of permeability and chronic inflammation.

Finally, following their production by B cell lineage cells in the lamina propria, dimeric immunoglobulin A (IgA) complexes bind to the immunoglobulin receptor (pIgR) on the basolateral membrane of IECs and are transported to the lumen [32]. This collaboration between immune cells and IECs provides an immune component to regulate commensal bacteria populations and contributes to intestinal homeostasis [33,34]. Indeed, several studies have shown that IgA binds colitogenic members of the microbiota [35], and that mice deficient in IgA or the receptor pIgR develop more severe colitis [36].

Intestinal barrier dysfunction has been directly linked to inflammatory disorders including IBD [5]. Marin et al. demonstrated that the tight junctions of Crohn’s patients are misaligned, fragmented, and severely disorganized [37]. Later, Gassler et al. reported the downregulation of junctional proteins (E-cadherin and α-catenin) and their mRNA in actively inflamed IBD patients [38]. In fact, intestinal barrier permeability is used as a prognostic indicator of relapse in patients with quiescent IBD [39,40]. In 2020, two novel studies demonstrated that intestinal barrier permeability can start years before the clinical diagnosis of IBD, suggesting the possibility that IBD can be prevented by early intervention; a brief review of these studies can be found here [41]. Immunologically, IBD results in impaired innate and adaptive immune responses. Specifically, high levels of T helper 17 cells lead to perpetuating disease. Furthermore, a decrease in Treg cells, a specialized subpopulation of T cells that act to suppress immune response, leads to defective anti-inflammatory mechanisms [42]. The role of GM metabolites in maintaining and synchronizing these main intestinal elements for continuous homeostasis and their downregulation in disease will be discussed here.

## 3. Oxygen Gradient Environment of the Mammalian Intestine

The mammalian intestine has evolved to function as a uniquely suited environment for the growth and survival of anaerobic microbes [43]. The GI tract harbors a distinct oxygenation profile [44] and is characterized by a high rate of metabolite circulation. Even at baseline, barrier epithelial cells that line the mucosa exist at a low-oxygen tension environment, defined as ‘physiologic hypoxia’ [45] (see Figure 1). Original studies revealed that the countercurrent oxygen exchange mechanisms of the GI tract provide for arterial blood supply diffusion to adjacent venules, along the crypt villus axis, resulting in graded hypoxia [46]. This steep oxygen gradient has been well documented in the distal colon of the GI tract, spanning from the anaerobic lumen, across the epithelium to the richly vascularized sub-epithelial mucosa [47]. Given the high energy requirement of the gut and the integral role of the epithelium in maintaining intestinal homeostasis, it is not surprising that these cells have evolved a number of mechanisms to cope with this austere metabolic environment [48]. During active inflammation, the combination of recruited leukocytes, edema, and vasculitis enhances the hypoxic gradient to become “inflammatory hypoxia” [45]. The microbiome plays a key role in maintaining this oxygen gradient, which is critical for nutrient absorption, barrier function, and immune responses in the intestine [49].

Tissue oxygenation has been observed using 2-nitroimidazole dyes (e.g., pimonidazole), a class of compounds known to undergo intracellular metabolism dependent on the level of tissue oxygenation [50] (Figure 1). These dyes were developed to image the low-oxygen environment of growing tumors [51] and have subsequently been used as tools to monitor levels of tissue oxygenation ex vivo. Nitroimidazoles can form covalent bonds with thiol groups of various tissue macromolecules only at a pO_2_ < 10 mmHg. Antibodies specific for these conjugated adducts provide a histochemical approach to estimation of tissue pO_2_. It is notable that this approach has not been established to titer tissue pO_2_; rather it is a qualitative estimate of tissue hypoxia (e.g., above or below pO_2_ of ~10 mmHg). In addition, studies in germ-free mice have revealed that such physiologic hypoxia results in large part to contributions from the microbiota [52] (see Figure 1).

Given the rather unique environment of the intestine, particularly the colon, a number of studies have shown that stabilization of the transcription factor hypoxia-inducible factor (HIF) in low-oxygen environments triggers the expression of genes that are essential to epithelial barrier function [53,54,55,56]. Additionally, HIF is one of the central regulators of overall tissue metabolism [57] and has profound influences on the inflammatory response [48]. HIF function is dependent on stabilization of an O_2_-dependent degradation domain (ODD) expressed on the α-subunit and subsequent nuclear translocation to form a functional complex with HIF-1β [58]. In normally oxygenated tissues, ferrous iron, 2-oxoglutarate (2-OG), and O_2_-dependent hydroxylation of two prolines (Pro564 and Pro402 of HIF-1α in humans) carried out by prolyl hydroxylase enzymes (PHD1-3) within the ODD of the alpha subunit initiates the association with the von Hippel-Lindau tumor suppressor protein (pVHL) and the recruitment of a ubiquitin-E3 ligase for degradation via proteasomal targeting [59,60]. Alternatively, in low-oxygen environments or chemically induced PHD inhibition, HIFα is stabilized. Recently, it was discovered that some metabolites play an indirect role in gut health through HIF stabilization. An in-depth review of the adaptive role of HIF in the context of barrier function has been reviewed in detail elsewhere [43].

## 4. Microbial Metabolites, Oxygen Metabolism and Barrier Regulation

The GM is a vital and permanent passenger represented in vast richness of microbial life inhabiting the gastrointestinal tract. The GM is composed of several species of microorganisms, including bacteria, yeasts, and viruses. Classified by phylum, Firmicutes and Bacteroidetes represent 90% of the gut bacterial composition, along other subdominant phyla such as Proteobacteria, Actinobacteria, and Verrucomicrobia [61]. *Lactobacillus*, *Bacillus*, *Clostridium*, *Enterococcus,* and *Ruminicoccus* are examples of the more than 200 different genera representing the Firmicutes phyla in the GM where *Clostridium* constitutes 95% of its composition. Bacteroidetes consists predominantly of *Bacteroides* and *Prevotella.* Less abundantly, the Actinobacteria phylum is mostly represented by the *Bifidobacterium* genus [62]. Similar to a fingerprint, everyone is born with a unique GM profile that will play a lifetime role in nutrient metabolism, maintenance of structural integrity of the gut barrier, energy, and immunomodulation. Established at birth and early life, GM composition can be directly impacted by factors including gestational age, delivery method, milk feeding, and weaning [63] as well as antibiotic use [62]. Likewise, the complex GM–host interaction begins at birth where the GM initiates immune development, while at the same time the host orchestrates GM composition [64]. The first years of life represent a critical timepoint where the establishment of a healthy GM and a respectful GM–host balance will regulate metabolism, immunity, and prevention of disease for life. A comprehensive report of the microbiota composition and individual variations thereof can be found here [62].

In healthy mammals, the host–microbiota interaction is symbiotic with regard to host immunity, energy metabolism, and cellular communication [29,65,66,67,68]. The GM is key in harvesting nutrients from the diet. GM metabolizes undigested food that safely reaches the large intestine (e.g., fiber and some starches, sugars). The metabolism of these dietary components yields many active microbial metabolites. Furthermore, the GM interacts with the epithelium in the local environment, which initiates the production of additional metabolites [69], facilitating a metabolite-mediated communication or “crosstalk” essential for gut health [70]. It is notable that shifts in GM composition and metabolite production may contribute to the development of mucosal diseases such as IBD [71,72]. These microbiota-derived metabolites include short-chain fatty acids, tryptophan catabolites, purines, secondary bile acids, and polyamines. The following section will discuss the role of selected metabolites in cell communication, energy balance, immunomodulatory activities, and gut barrier regulation.

### 4.1. Short-Chain Fatty Acids (SCFAs)

Dietary fibers (DF) are important carbohydrates that the human body is unable to digest. These fibers pass through the small intestine into the colon, where they are fermented by the GM. Pectin, inulin, α-glucans, oligosaccharides, and guar gum are fermentable fibers that constitute the main energy source for GM [61]. The distribution of consumed DF and resultant fermentation can shape the diversity and function of the GM [73], influencing the production of microbiota-derived metabolites. High DF intake correlates with reduced disease incidence and death [74,75,76]. It has been shown that DF are important for homeostatic gut barrier function [77,78], whereas the lack of DF intake is associated with autoimmune disease and IBD development [79,80]. These positive effects are attributed to microbiota-derived metabolites, notably SCFAs, interacting with intestinal cells through various mechanisms. Briefly, SCFAs are a requisite waste product to balance redox equivalent production in the anaerobic gut lumen. In the body, SCFAs are classified as saturated aliphatic carboxylate salts between one and six carbons in length—formate, acetate, propionate, butyrate, valerate, and hexanoate. Acetate, propionate, and butyrate are the most abundant, comprising > 95% of SCFAs, and exist in a molar colonic ratio of approximately 60:20:20, with total SCFAs reaching 140 millimolar (mM) in the proximal colon and 70 mM in the distal colon [81]. The majority of SCFAs are rapidly absorbed by colonocytes, with only 5–10% secreted in feces. These SCFAs have a significant impact on host physiology as energy substrates, gene expression regulators, and signaling molecules for specific receptors [82,83,84,85].

#### 4.1.1. SCFAs, Transporters, and Anti-Inflammation

Carrier-mediated transport represents the most important route of entry of SCFAs in their anionic form into the colonic epithelium [86]. Several transport systems including monocarboxylic transporter 1 (MCT1) and 4 (MCT4) operate in cellular uptake of SCFAs. Specifically, sequestration of butyrate to the colon is due in large part to the different affinities of the apical (K_m_ = 1.5 mM) and basolateral (K_m_ = 17.5 mM) SCFA–HCO_3_^-^ exchange transporters, which confine butyrate to colonocytes [87,88] (Figure 2). Similarly, the affinity of the apical MCT1 transporter for butyrate is also higher than the basolateral transporter MCT4, and the higher intracellular pH renders all SCFAs in the dissociated form, which limits passive diffusion across the basolateral membrane. In line with butyrate production, MCT1 is mostly observed in the colon and it is considered to be the primary transporter of butyrate [89]. Peripheral systemic availability of microbiota-derived butyrate has been shown to be less than 2%, as the vast majority of butyrate is utilized by colonocytes [90].

SCFAs exert anti-inflammatory influences in the intestinal mucosa in part by activation of G protein-coupled receptors (GPCRs) present in IECs and immune cells. GPCR41 and GPCR43 play a role in the immune surveillance of the colonic mucosa providing communication between SCFAs and mast cells [91]. Butyrate reduces lipopolysaccharide-induced NF-κB activation via GPR109A acting as a tumor suppressor [92]. Additionally, GPR43-bound acetate promotes potassium efflux and hyperpolarization in colonic cells, leading to NLRP3 inflammasome activation [93]. NLRP3 inflammasome activation leads to the release of proinflammatory cytokines (e.g., IL-1β and IL-18). The current body of knowledge in the relationship of these receptors and SCFAs has been reviewed elsewhere [94]. Furthermore, butyrate directly regulates immune cells, such as macrophages [95], dendritic cells [96], lymphocytes [95], and inhibits cytokines (e.g., IL12-p70 and IL-23) [97]. Lastly, as it is well known, butyrate is a potent histone deacetylase (HDAC) inhibitor [98] (Figure 2), affecting many processes including the inflammatory response. HDACs regulate innate immunity pathways, such as myeloid cell differentiation, TLR and IFN-mediated inflammatory response [99]. This demonstrates the importance of the localization of SCFAs in the colon and all the interactions with receptors, immune cells, and enzymes to regulate and coordinate proper immune function and intestinal health.

#### 4.1.2. SCFAs as the Primary Colonocyte Energy Source

Another aspect of butyrate sequestration is that it is the preferred energy source of the colonic epithelium, with oxidation of this SCFA accounting for over 70% of cellular oxygen consumption in the distal colon [100]. Colonocytes preferentially utilize butyrate over acetate and propionate, where it is oxidized to ketone bodies and CO_2_. Greater than 95% of microbiota-derived butyrate is utilized by colonocytes for energy. An energy deprived state (monitored by a decrease in enzymes involved in the tricarboxylic acid cycle) results in lower ATP levels, and ultimately increased autophagic flux, in colonocytes from germ-free (GF) mice. Recolonization of GF mice with butyrate-producing bacteria (mostly of the Firmicutes phylum) or butyrate treatment of GF colonocytes increases oxidative phosphorylation and returns autophagy to baseline [89,101]. As an energy substrate, butyrate undergoes β-oxidation to form acetyl-CoA, which enters into the tricarboxylic acid cycle (TCA) to produce the reducing factors that drives the electron transport chain (ETC) and oxygen consumption to ultimately regenerate ATP. Maintenance of the mucosal barrier requires cytoskeleton stability, requiring substantial energy pools, and the rapid utilization of butyrate not only prevents butyrate from escaping into systemic circulation but also provides the requisite energy for IECs to rapidly polarize and form strong AJCs [102].

#### 4.1.3. Butyrate and the Epithelial Barrier

Multiple studies have shown that the selective sequestration of butyrate in the colon influences intestinal barrier function. For instance, an mRNA-based screen of intestinal epithelial cells exposed to physiologic concentrations of butyrate revealed the repression of claudin-2 (*CLDN2*), a “leaky” claudin that increases permeability. The mechanisms of butyrate activity were subsequently traced to induction of the IL-10 receptor (IL-10Rα) and IL-10-enhanced protein expression on IECs through HDAC inhibition [103]. Other studies have shown butyrate to induce the expression of “sealing” TJ proteins such as CLDN1, also through HDAC inhibition [104]. More recently, a newly characterized gene of interest to barrier function, synaptopodin (*SYNPO*) was identified via a single-cell RNA sequencing approach (scRNAseq). SYNPO is an important intestinal epithelial tight junction protein. Like claudins, the mechanisms of butyrate regulation of SYNPO were via HDAC inhibition (Figure 2). SYNPO was additionally shown to promote wound healing as an important component of the butyrate-derived wound healing response in vivo [105]. Altogether, butyrate can coordinate the repression of a “leaky” TJ protein, the induction of “sealing” TJ proteins, and coordination of wound healing responses through HDAC inhibition. While HDAC inhibition impacts the expression of ~2% of mammalian genes [98], butyrate regulates barrier function by targeting multiple genes through HDAC inhibition. Additionally, SYNPO was induced at the protein level after 6 h of butyrate treatment, while other studies showed that CLDN2 levels were reduced by butyrate after 24 h, which corresponded with peak IL-10Rα induction at 24 h, followed by a CLDN1 increase after 36 h [103,104]. The observations suggest that butyrate may temporally orchestrate the induction and repression of specific TJ proteins to ultimately promote barrier function. Wang et al. showed in their scRNAseq that multiple TJ and actin-associated genes were upregulated by butyrate including cingulin (*CGN*), *CLDN1*, *CLDN3*, as well as genes related to cellular motility including *MYLIP* and *KIF11* [105] (Figure 2). This suggests that butyrate simultaneously and purposefully influences multiple genes related to these processes as, naturally, the AJC is comprised of many components and active epithelial restitution to begin wound healing requires many proteins working in concert.

#### 4.1.4. Butyrate-HIF Regulation of Epithelial Barrier

Butyrate-producing bacteria are anaerobic, taking advantage of the unique oxygen environment in the intestine with a low O_2_ concentration in the lumen. Furthermore, the metabolism of butyrate in the epithelium further reduces the already low levels of O_2_ to the point of HIF stabilization (a transcription factor involved in barrier protection) [106]. Accordingly, mice lacking microbiota-derived butyrate (e.g., GF mice) show diminished hypoxia localization (Figure 1) and reduced HIF stabilization at baseline [52]. Recently, an alternative butyrate-HIF stabilization mechanism independent of β-oxidation was reported [107]. Analysis using a combination of recombinant HIF prolyl hydroxylase (PHD) enzyme and 1D-NMR experiments found that butyrate stabilizes HIF by acting as a direct, non-competitive inhibitor of HIF PHD in vitro and in vivo. Interestingly, shorter or longer SCFAs (e.g., acetate, propionate, and valerate) exhibited a much higher PHD IC_50_ and no binding according to NMR. This implicates butyrate as a significant and dynamic endogenous regulator of HIF in IECs [107].

HIF contributes to epithelial barrier function in a number of ways. Butyrate induces barrier function in vitro (measured by FITC-dextran flux) but not in IEC monolayers lacking HIF1α, indicating a fundamental role for HIF in maintaining barrier [52]. Selective knockdown of HIF-1α in murine IECs demonstrated major defects in mucosal barrier integrity. This could partly be due to HIF-1α directly regulating the expression of CLDN1, a crucial TJ protein [108]. HIF-1α also upregulates MUC2, the major component of the mucus layer, as well as human β-defensin 1 (HBD-1), which is the only constitutively secreted antimicrobial peptide in the intestine [109,110]. Likewise, HIF is a transcriptional regulator of intestinal trefoil factor (i.e., TTF3), a 40 amino acid lectin protein that binds and crosslinks mucins [53]. HIF-2α regulates creatine kinases and the creatine transporter that co-localize with AJs and supply energy at junctional sites for tasks such as tight junction assembly, maintenance, and restitution [111,112]. Additionally, HIF-1α regulates the expression of ectonucleoside triphosphate diophosphohydrolase 1 (CD39) and 5′-nucleotidase (CD73), which enzymatically convert adenosine triphosphate (ATP)/adenosine diphosphate (ADP) to adenosine monophosphate (AMP) and AMP to adenosine, respectively. Adenosine signaling plays a key role in the perfusion of the intestinal mucosa and promotes intestinal barrier function through activating the adenosine 2B receptor (A2BR), which is highly expressed in the intestinal mucosa and is transcriptionally regulated by HIF-1α [113].

#### 4.1.5. SCFAs in Disease

Dysbiosis in IBD patients is characterized by loss of GM diversity, with higher abundance of Proteobacteria and loss of SCFAs/butyrate-producing bacteria mainly of the Firmicutes phylum. Specific studies show that a decrease in *F. prausnitzii*, a butyrate-producing bacteria is a hallmark of active IBD patients [114,115], ultimately resulting in lower luminal concentrations of butyrate. Multiple studies, for example, have revealed lower concentrations of luminal butyrate levels and reduced overall abundance of butyrate-producing organisms in patients with IBD (e.g., certain *Roseburia* and *Faecalibacterium* genera) [116,117,118]. While not conclusive, this correlative observation is supported by data demonstrating that specifically increasing fecal butyrate levels diminishes intestinal inflammation and some symptoms in patients with ulcerative colitis [119]. A recent meta-analysis of 25 studies of patients with ulcerative colitis suggested a nearly 65% clinical response rate to fecal microbiota transplant (FMT) that corresponded with a broad increase in microbial diversity including butyrate producers (e.g., Firmicutes and *Clostridium clusters IV*, *XIVa*, *XVIII)* [120].

Furthermore, among the abnormalities observed in IBD patients, it has been found that MCT1 protein expression (butyrate transporter) and the transcript of its encoding gene *SLC16A1* are reduced in inflamed mucosa [121,122,123]. In fact, an important inverse correlation is observed between butyrate uptake/oxidation and the Mayo endoscopic subscore and Geboes histological score [121]. Specifically, genes encoding enzymes responsible in butyrate oxidation/uptake are downregulated in inflamed mucosa of IBD patients [121,122,123]. Additionally, inflammatory cytokines inhibit butyrate uptake, oxidation (e.g., TNF-α), and MCT1/*SLC16A1* expression [124,125,126]. These findings suggest that inflammation is directly related to SCFAs (specifically butyrate) synthesis, uptake, and metabolism, suggesting that SCFAs supplementation alone may not be sufficient to regain homeostasis. Promising combined clinical approaches utilizing butyrate and butyrate-producing bacteria have resulted in prevention of relapse in IBD patients [127,128]. Collectively, these findings demonstrate the importance SCFAs in gut health and that a combination of approaches to regain butyrate uptake and metabolism may be a promising treatment in the clinical management of IBD.

### 4.2. Indoles and Indole Derivatives

#### 4.2.1. Biosynthesis of Indoles

1*H*-indole (hereafter, “indole”) and indole derivatives are gut bacterial metabolites resulting from degradation of dietary tryptophan [129]. Such derivatives, for example indole-3-propionic acid (IPA), indole-3-pyruvate (IPy), and indole-3-acetic acid (IAA), resemble indole except for substitution at the 3 position with a variety of diverse moieties. Generation of indole from tryptophan occurs through a single-step deamination reaction (EC 4.1.99.1), which generates pyruvate and ammonia as additional products [130]. The enzyme responsible for this reaction, TnaA (tryptophanase), has been described in diverse bacteria including both Gram-positive and Gram-negative species with varying degrees of conservation to the prototypic *E. coli* K-12 sequence [131]. Generation of substituted indoles from tryptophan occurs through a more complicated process often unique to the synthesizing organism; for example, the obligate anaerobe *Clostridium sporogenes* generates IPA through a multi-step pathway involving the production of IPy, indole-3-lactic acid (ILA), and indole-3-acrylate (IA) as immediate precursors along with synthesis of IAA under certain circumstances [132]. Similar multi-step biosynthetic pathways have been described for other bacteria, highlighting the diversity of microbial tryptophan metabolism (reviewed in [133]). Notably, indole and indole derivatives have been shown to be exclusively derived from the gut microbiota: studies using germ-free mice and mice treated with broad-spectrum antibiotics have shown a loss of indoles in the absence of an intact GM, indicating an essential role for intestinal microbiota in the metabolism of dietary tryptophan [132,134,135]. Indoles are abundant in the gut, and indole has been consistently measured in the millimolar range in human fecal extracts [134,135]. In addition to the influences on the mammalian intestinal tract (discussed in further detail below), indoles have been characterized as intercellular signaling molecules, controlling a variety of cellular processes including virulence gene expression, sporulation, plasmid stability, cell cycle control, and biofilm formation [136,137,138,139,140]. The ability of indoles to modulate eukaryotic cell function implies that these molecules act as interkingdom signaling molecules and suggests a co-evolution between metazoans and their gut symbionts [141].

#### 4.2.2. Protective Actions of Indoles

The protective influence of indoles on the mammalian gastrointestinal tract has been known since at least 2010, when Bansal et al. described the induction of tight junction genes by indole treatment in the colorectal adenocarcinoma cell line HCT-8 [142]. Bansal et al. further demonstrated that acute treatment with indole enhances HCT-8 barrier function, as measured by transepithelial electrical resistance (TEER), and is protective during TNF-α-induced inflammation by suppressing nF-κB signaling and IL-8 secretion, while inducing expression of the anti-inflammatory cytokine IL-10. These results were corroborated by subsequent studies that demonstrated a positive influence of indoles on barrier function in Caco-2 and T84 colorectal adenocarcinoma cell lines, including protection against TNF-α-induced barrier dysfunction, as measured by both TEER and flux of the fluorescent tracer FITC-dextran across the cellular monolayer [143,144,145]. IPA was further shown to be protective in the dextran sodium sulfate (DSS) animal model of colitis, reducing tissue pro-inflammatory cytokines and ameliorating intestinal histopathology [145]. The protective benefits of indole derivatives have been associated with stimulation of aryl hydrocarbon receptor (AhR) signaling (Figure 3), as studies have indicated that the protective benefits of indoles are lost when AhR is knocked out, pharmacologically inhibited, or its signaling obviated due to loss of its nuclear binding partner ARNT [144,145]. AhR is a ligand-dependent transcription factor expressed in diverse cell types that binds a wide variety of both xenobiotic factors (e.g., dioxins) and endogenous ligands, resulting in activation of an anti-inflammatory/pro-barrier transcription program (reviewed in [146]). Activation of AhR by endogenous ligands, such as the host-derived tryptophan metabolite kynurenine, has been shown to induce expression of IL-10R1 and promote wound healing following intestinal insult [147]. Similar observations have been made for indoles: treatment of intestinal epithelial cells with indole derivatives induces IL-10R1 in vitro, and mice treated with IPA show enhanced colonic IL-10R1 expression during DSS colitis [145]. Additionally, indole-dependent AhR signaling was found to prevent inflammation-induced activation of the actin-regulatory protein ezrin, a protein implicated in loss of apical junction complex integrity, as well as activation of myosin light-chain kinase, which was similarly involved in regulation of epithelial barrier integrity [144,148] (Figure 3).

Although the importance of AhR signaling in mediating the pro-barrier influences of indoles has been demonstrated by several studies, evidence suggests that the pleiotropic influences of indoles maintain epithelial homeostasis through diverse mechanisms. One study found that indoles, namely indole and IPA, act synergistically upon the human pregnane X receptor (PXR) to mitigate tissue histopathology and pro-inflammatory cytokine expression in the indomethacin model of small intestine inflammation [149]. PXR is a transcription factor that can bind a diverse array of ligands and has been implicated in attenuating intestinal inflammation through suppressing nF-KB target gene expression [150]. The importance of PXR in maintaining epithelial homeostasis was found to be dependent on Toll-like receptor (TLR) signaling: mice deficient in both PXR and TLR4 (which recognizes lipopolysaccharide) did not show the increased susceptibility to indomethacin evidenced by mice deficient in PXR alone. These findings suggest that the intestinal epithelium exists in a finely tuned state between pro- and anti-inflammatory responses, and disruption of one pathway may elicit deleterious influences by disturbing this balance.

Indoles have also been shown to be protective in the context of intestinal inflammation by acting on the innate immune system. For example, indoles, including indole, IPA, and IAA, have been demonstrated to inhibit neutrophil myeloperoxidase (MPO) in assays using both purified, recombinant human MPO and primary human neutrophils [151,152]. MPO, through its generation of hypochlorous acid from hydrogen peroxide, is essential for neutrophil anti-microbial activity but can cause damage to “bystander” tissues during uncontrolled inflammation [153]. Indole and IPA have been found to bind MPO directly to block the generation of HOCl, suggesting a role for regulation of innate immune processes by gut microbial metabolites [151].

#### 4.2.3. Indole-Derived Gut Microbiota Shifts

Indoles may regulate intestinal homeostasis through modulating the identity of the gut microbiota. One study demonstrated that exogenous supplementation of indole significantly reduced gut microbiota perturbation induced by indomethacin, correlating with less severe tissue histopathology and lower fecal calprotectin (a metric of intestinal inflammation) [154]. Other studies have observed loss of intestinal indole metabolites during active inflammation, including in samples from IBD patients, suggesting a link between gut indoles and disease progression [145]. Similarly, treatment of mice with indole-3-carbinol (IC) during 2,4,6-trinitrobenzenesulfonic acid-induced (TNBS-induced) colitis prevented a pathologic shift in the gut microbiota and ameliorated disease in a manner dependent on IL-22, a member of the IL-10 cytokine family [155]. Indole has also been implicated in the pathogenesis of *Clostridioides difficile* and has been determined to be a predictor of *C. difficile* disease relapse [156,157]. In this way, future studies should investigate the potential of indole and indole derivatives as potential therapies for intestinal inflammatory disorders through their ability to directly modulate barrier function, inflammatory responses, and the identity of the gut microbiome.

### 4.3. Hypoxanthine and Other Purines

Intestinal healing and barrier repair require sufficient nucleotide generation to provide RNA for protein transcription and DNA for proliferation. Furthermore, high levels of ATP are needed to maintain the energy balance necessary to drive cytoskeletal function for wound restitution and endoplasmic reticulum function for mucin synthesis and secretion [158]. Since 1998, it has been known that adenine nucleotides (specifically ADP and ATP) significantly stimulate IEC migration and enhance structural and functional regeneration in vivo [159], indicative of the fundamental role nucleotides and associating metabolism play in eukaryotic cellular function.

#### 4.3.1. Hypoxanthine as a Biological Marker of Intestinal Barrier

Pursuant to understanding intestinal barrier-related adenylate energy flux, Lee et al. developed an HPLC-based profiling method to monitor changes in high-energy phosphates and adenylate metabolites [160]. In this study, they elucidated the role of hypoxanthine (Hpx, a naturally occurring purine) as a checkpoint metabolite in IEC function, with Hpx promoting cellular energetics to the benefit of cytoskeletal and barrier function. A “calcium switch” experiment [161] disrupts epithelial tight junctions and results in a >90% loss of barrier that recovers over time as monitored by TEER. Surprisingly, Hpx levels had an inverse correlation with barrier integrity, with a 3.5-fold increase in Hpx at the lowest barrier function. This suggests that Hpx is a marker of the rapid adenylate metabolite pool regulation by IECs during stress. Hpx supplementation studies demonstrated an enhanced barrier formation rate and an epithelial monolayer with significantly higher resistance in Hpx-treated cells. Hpx significantly accelerated wound closure rate, demonstrating a role in improving cellular migration. Additionally, it was discovered that Hpx increases total available epithelial cellular energy, as seen in an increase in cellular phoshocreatine (PCr) and ATP. Alternatively, inhibition of adenylate metabolite flux through Hpx significantly impacted barrier development, altogether identifying an important role for the purine salvage pathway in IEC function. A metabolomic screen from healthy and colitic murine colon tissue revealed a >65% decrease in Hpx during active inflammation. Furthermore, the loss of Hpx strongly correlated with disease markers (e.g., weight loss, colon length) [160].

Given the unique environment of physiological hypoxia at the intestinal epithelial–luminal interface, the role of Hpx in barrier development and energetics was explored in hypoxia. IECs were exposed to physiologically relevant hypoxia (1% O_2_, 40 h) in the presence and absence of Hpx. Controls suffered a significant drop in barrier strength as shown by TEER, with Hpx supplementation wholly preventing the hypoxia-induced loss of barrier function. Furthermore, hypoxia incurred a significant loss of energy pools, while Hpx supplementation salvaged this energy loss by increasing both PCr and ATP levels [160].

#### 4.3.2. Hypoxanthine Is Also a GM Metabolite

The de novo synthesis of purines in IECs is limited due to its high energy cost, requiring 5 ATP molecules for production of one purine, in an environment of energetically-depleting hypoxia. Thus, purine metabolism typically depends on the salvage of exogenously supplied purine substrates for nucleotide biogenesis [162]. The purine salvage pathway functions to utilize nucelobases such as Hpx for the biosynthesis of ATP and equilibrating energy. A recent report sought to explore sources of purines in the murine mucosal environment. Surprisingly, water-soluble extracellular fecal metabolite analyses of conventionally raised (CR) mice revealed significant concentrations of salvageable and available purines, mostly in the form of Hpx and xanthine (Xan) (Figure 4). In contrast, fecal extracts from streptomycin-treated mice exhibited a 90% reduction in purines, identifying the microbiota as a distributor of microbiota-sourced purine (MSP) available for salvage by the intestinal epithelium [158].

To assess the importance of MSP in metabolism and barrier function, control GF mice (GF-CNTL) were monocolonized with purine-producing *E. coli* K12 (GF-K12), and then both were subjected to DSS colitis. In contrast to GF-K12 mice, which showed minimal signs of disease, GF-CNTL mice demonstrated notable weight loss and significant mortality during the course of treatment. Through the salvage of MSP, healthy GF-K12 mice showed significant increases in purine levels versus GF-CNTL, including Hpx, Xan, Inosine (a hypoxanthine-derived nucleoside) and high-energy phosphates. Alternatively, DSS-induced colitis diminished most tissue purines in GF-K12 mice but without loss of ADP and with increases in ATP and PCr. This result suggests that colonic tissue is reliant upon MSP in order to maintain adenylate nucleotide levels and energy balance during insult. Furthermore, Hpx supplementation in streptomycin-treated mice (which show loss of MSP) subjected to DSS-induced colitis proved to be protective by reducing weight loss and colon shortening, as well as increasing tissue adenylate energy levels similar to K12-colonized mice [158]. Thus, purines (especially hypoxanthine) seem to play a vital role in intestinal barrier health in a manner tightly correlated with energy balance. Further exploration of purines in IBD patients may shine light on promising novel treatments.

### 4.4. Secondary Bile Acids

#### 4.4.1. Secondary Bile Acids Metabolism

Primary bile acids (PBAs), specifically cholic acid (CA) and chenodeoxycholic acid (CDCA), are synthesized from cholesterol in the liver. PBAs are constitutively produced and, along with cholesterol and phospholipids, comprise the gallbladder-stored bile salts. Following a meal, bile salts are secreted into the small intestine. Prior to their secretion, CA and CDCA are conjugated to glycine or taurine via an amide bond [163]. This transformation allows the formation of micelles [164], which in turn emulsify lipids and fat-soluble vitamins for absorption. After digestion, approximately 95% of the PBAs are reabsorbed by the distal ileum and recycled from circulation by the liver [165]. The remainder that evade absorption reach the colon and dynamically interact with the microbiota, where they are metabolized into secondary bile acids (SBAs) produced solely by the GM.

Numerous SBAs are produced by microbial metabolism, including litocholic acid (LCA) and deoxycholic acid (DCA). A small population of intestinal species in the Firmicutes phylum and genus *Clostridium*, including *C. scindens*, *C. hiranonis*, *C. hylemonae*, and *C. sordelli* are capable of producing SBAs [166]. The biochemistry of this metabolism is particularly interesting, broadly encompassed by two steps: hydrolysis of the amide bond for deconjugation via bile salt hydrolases (BSH) producing free primary bile salts; followed by transformation to secondary bile acids, mainly by 7α-dehydroxylation reactions [166]. It appears that the latter reaction is restricted to free bile acids, thus the deconjugation step is a prerequisite [166]. The conversion of PBAs to SBAs by 7α-dehydroxylases is considered one of the most physiologically relevant microbial transformations in the body [166]. In addition, SBAs undergo selective microbial isomerization giving them different immunomodulatory properties, for example isoDCA (an isomer of DCA) promotes regulatory T cells (Treg) differentiation, while DCA itself cannot [167]. This suggests a possible on-demand supply of different SBAs depending on the current intestinal environment.

#### 4.4.2. Secondary Bile Acids and Receptors

SBAs are potent nuclear receptor ligands, binding to farnesoid X receptor (FXR), vitamin D receptor (VDR), PXR, and act as endogenous agonists for the microbial G protein coupled-bile acid receptor (TGR5) [61]. These receptors play an important role in many cells, including IECs and immune cells, with response to endogenous and bacterial antigens. FXR regulate intestinal immune responses driven by the GM with altered bile acids profiles during dysbiosis [168]. Intestinal diseases such as IBD downregulate bile acids biotransformation genes mostly from the Firmicutes phylum (a.k.a main butyrate-producing bacteria), resulting in low SBAs and high PBAs levels compared to healthy individuals [169,170]. Various in vivo colonic models including FXR null mice [171], downregulation of FXR [172] and antibiotic-induced GM alterations [173] resulted in uncontrolled intestinal inflammation and premature death. Obeticholic acid (INT-747) is a semi-synthetic SBA and a potent FXR receptor agonist. In human monocytes and dendritic cells cultured in vitro, the expression of key inflammatory cytokines and chemokines are inhibited by INT-747-mediated FXR activation [174]. Likewise, the expression of pro-inflammatory genes is also repressed by FXR activation in IECs [175]. In vivo, INT-747-mediated FXR activation improves epithelial barrier integrity while decreasing inflammatory cytokines (e.g., IL-1β, IL-6) and CCL2 chemokine in DSS or TNBS-induced colitis murine models [174,175].

Furthermore, SBAs activate TGR5, unleashing a plethora of intestinal immunomodulatory and anti-inflammatory influences, mainly in macrophages and monocytes [176]. Briefly, TGR5 regulates energy metabolism and glucose homeostasis. In addition, pro-inflammatory mediators such as, IL-1, IL-6, and TNF-α are inhibited by SBA-mediated activation of TGR5 [177], while the intestinal anti-inflammatory cytokine IL-10 levels increase [178]. In vivo experiments discovered that downregulation of TGR5 results in destroyed architecture of epithelial tight junctions and abnormal distribution of zonulin-1 [179].

Lastly, VDR downregulation and vitamin D deficiency are common biomarkers in patients with IBD [180]. Using VDR as a receptor, bile acids (mainly LCA) control various physiological processes, including cell differentiation and inflammatory resolution. In various colitis models, VDR knockout mice displayed low levels of antimicrobial peptides, barrier dysregulation, and increased mortality [181,182]. Additionally in colitis models, mice with transgenic human VDR in IECs exhibit a mucosal barrier protection mechanism as elucidated by preserved TEER, reduction in IEC apoptosis, caspase-3 deactivation, and downregulation of p53-upregulated modulator of apoptosis (PUMA), an important inducer of IEC apoptosis in IBD [183].

Interestingly, bile acids play a major role in GM composition. Kakiyama et al. reported a relationship between liver health, GM populations, and fecal bile acid profiles [184]. In this study, it was observed that bacterial dysbiosis is linked to lower levels of bile acids reaching the intestine in patients with early and advanced cirrhosis as compared to control patients. This GM dysbiosis was elucidated by a significant reduction in commensal Gram-positive members (e.g., *Blautia and Rumminococcaceae*) and an increase in pro-inflammatory and harmful taxa *Enterobacteriaceae*, accompanied by decreased fecal bile acid levels [184,185]. Alternatively, a diet rich in bile acids (CA) resulted in phylum-level shifts of the GM, with Firmicutes populations vastly expanding in rats [186]. As a possible mechanism, bile acids act upon certain bacterial membranes due to their hydrophobicity and detergent properties exhibiting direct antimicrobial effects. A full report of the role of bile acids in GM composition can be found here [187].

Overall, SBAs play a pivotal role in the activation of important receptors that control inflammation and immunity. SBAs are broadly responsible for the regulation of Tregs, monocytes, and macrophages in the intestinal barrier. Furthermore, they influence GM composition in health and disease. These vital functions elicit a tremendous interest in further exploration of natural SBAs or novel synthetic derivatives as therapeutic options in inflammatory diseases.

### 4.5. Polyamines

Natural aliphatic biomolecules containing at least two amine groups are termed polyamines (hereafter PAs). Their carbon backbone allows them to bond to hydrophobic molecules, and their amino groups (usually carrying a positive charge under physiological pH) allow them to bind to anionic moieties. In mammalian cells, the total concentration of PAs is in the mM range. However, free intracellular PA concentration is much lower (approx. <10%), due to the fact that most of these cationic moieties are constantly binding to negatively charged molecules, including nucleic acids, proteins, and phospholipids, often modulating their function [188]. It is generally known that PAs are mostly endogenous (de novo biosynthesis, catabolism); however, the microbiome is known to be a major source of luminal polyamines. The major PAs in mammalian cells are spermidine, spermine, and their precursor putrescine [189]. PAs are essential for various cell functions from cell proliferation and viability, immune system and apoptosis [190,191,192], derived mostly from the two amino acids ornithine and methionine, and largely regulated by two enzymes ornithine decarboxylase (ODC) and S-adenosylmethionine decarboxylase (SAMDC). Studies to date support the importance of polyamines for normal gut mucosal growth and barrier function [193,194,195]. Increasing the concentration of PAs stimulates gut mucosal renewal and enhances barrier function. Alternatively, inhibiting the major PA enzymes (ODC and SAMDC) leads to compromising the gut epithelia and barrier dysbiosis [193,196]. Furthermore, evidence that PA levels are remarkably impacted in active patients with IBD [194] and other mucosa-related diseases [195] has elicited recent interest in these biomolecules as a potential novel therapeutic approach.

PAs play various important roles in intestinal health, including epithelial renewal, barrier maintenance, and immunity. Reports indicate that a sustained supply of PAs to actively dividing cells is required for normal intestinal epithelial renewal and mucosal healing [193,197]. Growth stimulation is possible through PA-mediated regulation of various genes encoding growth-promoting proteins (e.g., *MYC*, *FOS*, and *JUN*) [198,199,200] and control of growth-inhibiting factors such as p53, nucleophosmin, JunD, TGF-β, and Smads [201,202,203,204]. Furthermore, PAs enhance intestinal barrier function by modulating intercellular junctions. PAs (mostly spermidine) are necessary for the expression of AJ constituent E-cadherin. It has been reported that E-cadherin stabilization is partially achieved through PA-induced increases in intracellular free Ca^2+^, a necessary cofactor for AJ formation [205]. E-cadherin expression is also due to enhanced gene transcription by activating MYC, which directly interacts with the E-pal box located at the E-cadherin promoter [206]. PAs regulate TJ protein Zonula occludens-1 (ZO-1) expression by modulating its gene transcription via *JUND* [207] and other TJs including ZO-2, claudin-2, and claudin-3, though the mechanisms of this regulation are still unknown [208]. PAs are also important in regulating immune responses. As an example, spermidine restored CD8^+^ T cell responses in elderly mice [209]. Additionally, PA spermine inhibits LPS-mediated production of nitric oxide and pro-inflammatory cytokines such as TNF-α, IL-1β, and IL-6 in macrophages and human mononuclear cells [210,211]. In the same context, biogenic amines such as spermine and histamine influence IL-18 secretion through inhibition of the NLRP6 inflammasome [212,213]. Gathering this data, it is clear yet again the important role that metabolites (including PAs) play in intestinal barrier restoration, wound healing, proliferation, and immunity.

## 5. Concluding Remarks

The vital role that the gut microbiota plays in intestinal health and systemic immunity is clear. The study of the complex machinery and mechanisms behind these benefits is likely in its infancy. Exploring further metabolite functions, receptors, metabolite-regulated transcriptional activity, and even novel metabolites opens a promising and vast field of research directed at ameliorating intestinal diseases. The study of microbial shifts and environment-dependent “on demand” biosynthesis of specific metabolites is also attractive.

Many questions remain; however, it is clear that metabolites interact with every cell type in the gut, from IECs to immune cells. Although a single approach (e.g., supplementation) might not be sufficient to restore barrier function, a combinatorial manipulation of receptors, metabolites, enzymes, and transcription factors might become a silver bullet—not forgetting that the simplest approach to procure a healthy gut is through a diet rich in necessary components for normal GM composition and metabolite biosynthesis.

## Figures and Tables

**Figure 1 cells-11-00944-f001:**
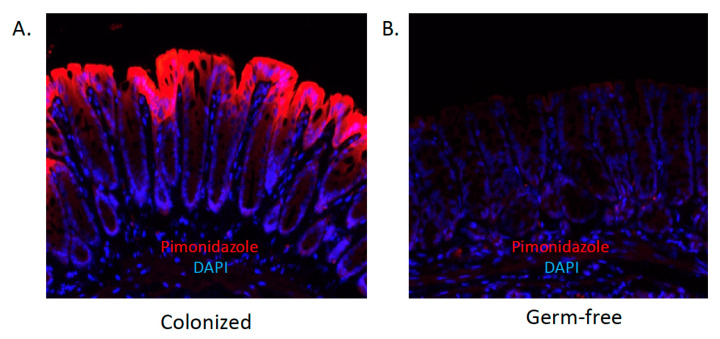
The contribution of the microbiota to “physiologic hypoxia”. Shown here are histologic sections of healthy colon derived from colonized (**A**) or germ-free (**B**) mice documenting low-oxygen regions (red) visualized by staining with pimonidazole, a dye that stains only in low-oxygen tensions. Nuclear counter-stain with DAPI is shown in blue. Note the near total lack of pimonidazole localization in the absence of microbial colonization.

**Figure 2 cells-11-00944-f002:**
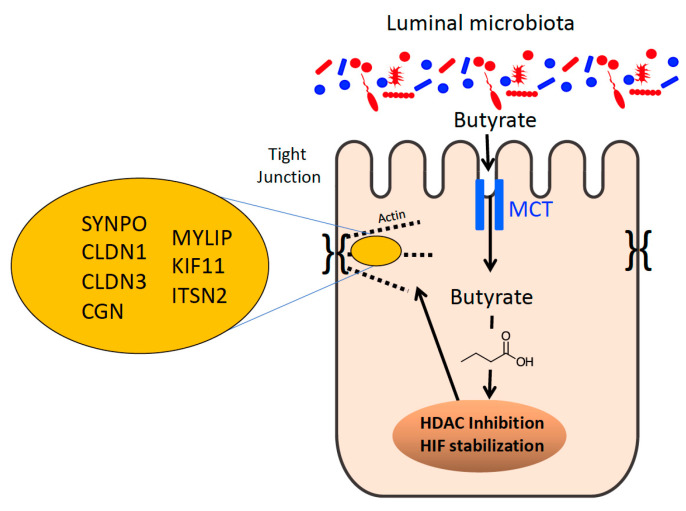
Microbial-derived butyrate regulates epithelial tight junction expression and function. Shown here is the influence of butyrate on tight junction protein expression. Through the actions of butyrate on HDACs and HIF, butyrate influences the expression of multiple TJ-associated proteins, including those indicated here. See text for further clarification.

**Figure 3 cells-11-00944-f003:**
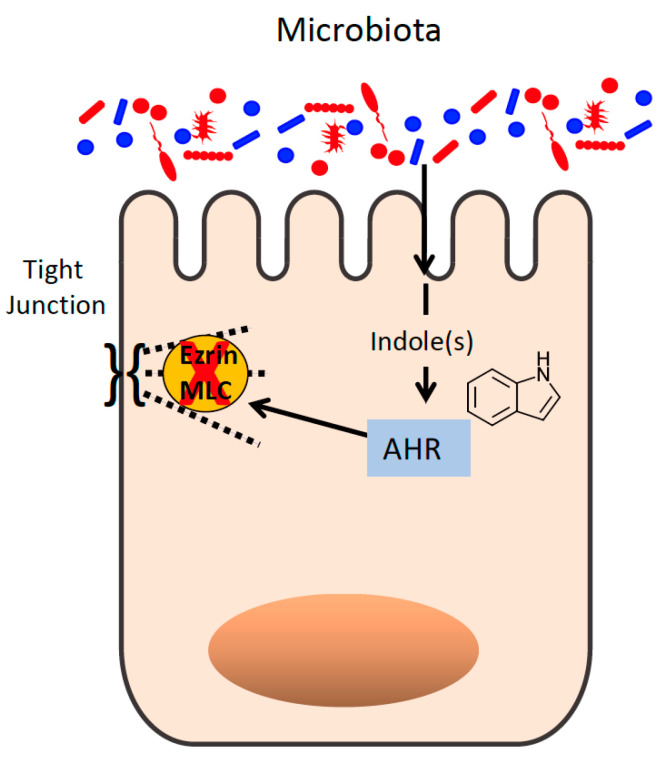
Microbe-derived indole regulates epithelial tight junction function. Indoles act through the aryl hydrocarbon receptor (AHR) to inhibit ezrin and myosin light-chain (MLC) kinase activity.

**Figure 4 cells-11-00944-f004:**
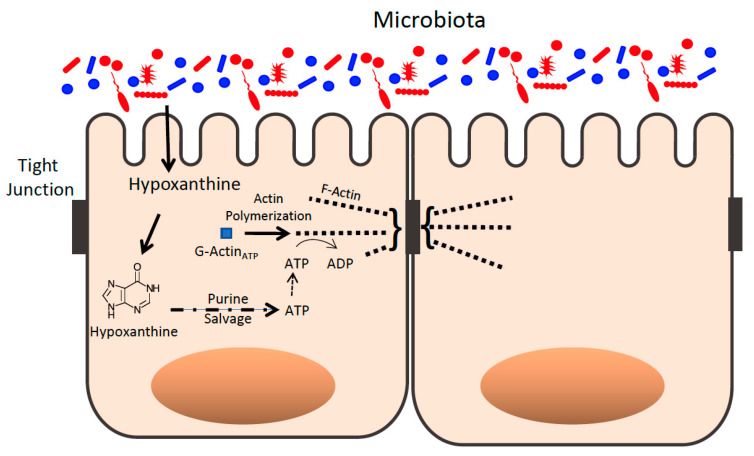
Purines derived from microbial sources influence the epithelial apical tight junction actin complex. Purines such as hypoxanthine are derived, in part, from microbial sources and are salvaged as templates for ATP generation that supports actin polymerization of G-actin to F-actin in the apical epithelial TJ complex.

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
