# Peer review of "Microbial Metabolite Regulation of Epithelial Cell-Cell Interactions and Barrier Function"

_cells, 2022, doi:10.3390/cells11060944_

Round 1

Reviewer 1 Report

The review from Ornelas et al aimed to summarize the role of gut microbial metabolites on gut barrier function. This is a timely review with growing evidence for an impaired gut barrier as a key mechanism by which the mutualistic relationship between the host and microbes is perturbed. This manuscript provided a good overview of the select groups of metabolites but lacked an in-depth review of the literature on molecular mechanisms. I have some concerns with the suitability of this manuscript being considered in this Special Issue – in various parts of the review I felt that the focus has drifted from barrier function to immune response and even IBD, although these were relevant to each other and might even be causatively related. My specific comments are as follows:

  1. The relationship between gut barrier function, immune response and IBD should be clearly described.
  2. Although the focus of this review is microbial metabolites rather than the gut microbiota per se, I strongly encourage the authors to include brief review of the relevant microbial producers wherever appropriate. Also, the description of gut microbiota in the current manuscript was too superficial, particularly the discussion of gut microbiota at the phylum level was too broad to provide any meaningful insights.
  3. The authors may wish to include brief review of data from gut microbiota-targeted interventions (animals or humans).
  4. I found the first paragraph of Section 4.1.1 (Line 226) out of place as it was on butyrate only, while the next paragraph was back on SCFAs in general.
  5. Lines 580-583: were the authors implying that IBD led to changes in butyrate producers and therefore a down-regulation of SBAs? Were then causative data to support this cascade of events?
  6. Line 615: what did the authors mean by “normal” in “normal gram positive members”?
  7. Lines 682-684: Please elaborate on “collaboration and overlap between metabolites” and “collegial sharing of receptors”.

Author Response

Thank you for your insight, we have addressed comments in a point-by-point fashion here.

1. The relationship between gut barrier function, immune response and IBD should be clearly described.

In the revised manuscript, paragraph describing this relationship and citing various reports was incorporated on page 4.

2. Although the focus of this review is microbial metabolites rather than the gut microbiota per se, I strongly encourage the authors to include brief review of the relevant microbial producers wherever appropriate. Also, the description of gut microbiota in the current manuscript was too superficial, particularly the discussion of gut microbiota at the phylum level was too broad to provide any meaningful insights.

Although the main focus of this review is on microbial metabolites and their role in regulation of cell-cell interactions and barrier function, microbial producers of these metabolites were briefly discussed throughout the paper. Section 4.1.5 describes the main SCFAs producing bacteria (Firmicutes phylum). Section 4.2.1 Biosynthesis of indoles, describes the generation of substituted indoles by Clostridium sporogenes. A brief sentence of bacteria capable of producing secondary bile acids was added at the end of page 16, Section 4.4.1.

In the revised manuscript, discussion of the gut microbiota composition was added on page 6 at the beginning of Section 4. A report on gut microbiota composition and individual variation was included for further inquiry.

3. The authors may wish to include brief review of data from gut microbiota-targeted interventions (animals or humans).

In the revised manuscript, we have included a section that discusses gut-microbiota targeted intervention in IBD (p. 10-11, section 4.1.5).

4. I found the first paragraph of Section 4.1.1 (Line 226) out of place as it was on butyrate only, while the next paragraph was back on SCFAs in general.

A brief comment about SCFAs transporters in general was incorporated in the first paragraph of Section 4.1.1. (page 8) of the revised manuscript.

5. Lines 580-583: were the authors implying that IBD led to changes in butyrate producers and therefore a down-regulation of SBAs? Were then causative data to support this cascade of events?

Referring to the following:

Intestinal diseases like IBD downregulate bile acids biotransformation genes mostly from the Firmicutes phylum (a.k.a main butyrate producing bacteria), resulting in low SBAs and high PBAs levels compared to healthy individuals.

The intention of the authors is to point out the indirect relationship between two different classes of metabolites (butyrate/SCFAs and secondary bile acid transformation) linked to the Firmicutes phylum.

6. Line 615: what did the authors mean by “normal” in “normal gram positive members”?

Replaced normal with commensal for clarity in the revised manuscript,.

7. Lines 682-684: Please elaborate on “collaboration and overlap between metabolites” and “collegial sharing of receptors”.

Because this comment is not strictly supported by existing literature, we have removed this comment from the revised manuscript.

Reviewer 2 Report

This manuscript provides an excellent summary of the interactions of cells and barrier functions in the intestine. It describes first the composition of the intestinal barrier and then the oxidative effect in eukaryotes, followed by the metabolites and their regulation. Thus, the manuscript aims precisely at the question of the influence of epithelial mucosal permeability.

All in all, this is a successful manuscript and there is only one small point that should be improved. In the introduction, the composition of the intestinal structure, barrier function and very nice defence functions are presented, but unfortunately the defence mechanism of the defensins was forgotten. The defensins, which are also called the „body's own antibiotics“, play a major role in Crohn's disease (Wehkamp et al. GUT 2004, 53(11)). It would be nice if this point could be integrated.

Author Response

Thank you for pointing this out. Indeed, defensins play a very important role in the innate host defenses. We have incorporated a brief paragraph about enteric defensins which can be found on page 3 of the revised manuscript. 

Reviewer 3 Report

  • “loss of barrier function is tightly linked to chronic inflammatory diseases, such as inflammatory bowel disease (IBD).”

Add a citation (for example: “Caviglia GP et al. Serum zonulin in patients with inflammatory bowel disease: a pilot study. Minerva Med. 2019 Apr;110(2):95-100. doi: 10.23736/S0026-4806.18.05787-7.”)

  • Use Oxford comma

  • Replace the term “flora” with microbiota or microbiome according to the specific setting

  • “SCFAs exert anti-inflammatory influences ….”

Acetate exerts anti- or pro-inflammatory effect?

  • Cite studies about the efficacy of butyrate in IBD (for example, “Caviglia GP et al. Efficacy of a Preparation Based on Calcium Butyrate, Bifidobacterium bifidum, Bifidobacterium lactis, and Fructooligosaccharides in the Prevention of Relapse in Ulcerative Colitis: A Prospective Observational Study. J Clin Med. 2021 Oct 26;10(21):4961.”. “Vernero M et al. The Usefulness of Microencapsulated Sodium Butyrate Add-On Therapy in Maintaining Remission in Patients with Ulcerative Colitis: A Prospective Observational Study. J Clin Med. 2020 Dec 4;9(12):3941.”)

  • “Clostridium difficile” -> Clostridioides

  • What about the possible role of obethicolic acid in IBD?

Author Response

  • loss of barrier function is tightly linked to chronic inflammatory diseases, such as inflammatory bowel disease (IBD).” Add a citation (for example: “Caviglia GP et al. Serum zonulin in patients with inflammatory bowel disease: a pilot study. Minerva Med. 2019 Apr;110(2):95-100. doi: 10.23736/S0026-4806.18.05787-7.”)

Thank you for the suggestion, citation was incorporated as seen on page 2 (citation 5) of the revised manuscript.

  • Use Oxford comma

Oxford comma incorporated throughout the document.

  • Replace the term “flora” with microbiota or microbiome according to the specific setting

Replaced flora with microbiota (page 2 and 11) of the revised manuscript.

  • “SCFAs exert anti-inflammatory influences ….” Acetate exerts anti- or pro-inflammatory effect?

According to various reports acetate exerts anti-inflammatory effects. Here are a examples of reports to support these findings:

  1. Xu, M., Jiang, Z., Wang, C. et al.Acetate attenuates inflammasome activation through GPR43-mediated Ca2+-dependent NLRP3 ubiquitination. Exp Mol Med 51, 1–13 (2019). https://doi.org/10.1038/s12276-019-0276-5
  2. Tedelind, S.; Westberg, F.; Kjerrulf, M.; Vidal, A. Anti-inflammatory properties of the short-chain fatty acids acetate and propionate: a study with relevance to inflammatory bowel disease. World journal of gastroenterology 2007, 13, 2826-2832, doi:10.3748/wjg.v13.i20.2826.

  • Cite studies about the efficacy of butyrate in IBD (for example, “Caviglia GP et al. Efficacy of a Preparation Based on Calcium Butyrate, Bifidobacterium bifidum, Bifidobacterium lactis, and Fructooligosaccharides in the Prevention of Relapse in Ulcerative Colitis: A Prospective Observational Study. J Clin Med. 2021 Oct 26;10(21):4961.”. “Vernero M et al. The Usefulness of Microencapsulated Sodium Butyrate Add-On Therapy in Maintaining Remission in Patients with Ulcerative Colitis: A Prospective Observational Study. J Clin Med. 2020 Dec 4;9(12):3941.”)

Citations and brief mention of clinical studies about efficacy of butyrate in IBD incorporated on page 10 of the revised manuscript.

  • “Clostridium difficile” -> Clostridioides

Changed to Clostridioides difficile (page 13 of the revised manuscript).

  • What about the possible role of obethicolic acid in IBD?

The role of obethicolic acid is discussed in page 16, section 4.4.2 of the revised manuscript.